# 3D printed sub-terahertz photonic crystal for wireless passive biosensing
Yixiong Zhao [1] ✉, Ali Alhaj Abbas[2], Masoud Sakaki [3], Gero Bramlage[4], Guillaume Delaittre[4], Niels Benson[3], Thomas Kaiser[2] & Jan C. Balzer [1] ✉

Monitoring pathogens has become a major challenge for society and research in recent years. Of great interest are refractive index sensors, which are based on the interaction between analytes and electromagnetic waves and allow label-free and fast detection. In addition, the electromagnetic waves can be exploited for wireless communication. However, current refractive index biosensors can only be read from a few centimeters. Here, we demonstrate an innovative concept of a passive wireless sensor based on a sub-terahertz photonic crystal resonator. The fabricated sensors have a reading range of up to 0.9 m and elevation and azimuth acceptance angles of around 90°. We demonstrate the stand-off detection of sub-μm thin-film proteins as test analytes. The proposed wireless sensor opens the door to a non-electronic, compact, and low-cost solution and can be extended to a wireless sensor network monitoring airborne pathogen, which may provide a pre-infection detection to prevent their spread efficiently.

The Covid-19 crisis has highlighted that viruses and other airborne pathogens pose a key challenge for the future. An important step in addressing this nanoscopic threat, in addition to developing an appropriate vaccine, was detection. Thus, an important milestone in containing the pandemic was the development and scaling of rapid tests based on pre-existing nucleic acid amplification methods[1]. However, these have the disadvantage that they allow detection only after infection. Consequently, to efficiently counter future pandemics, the development of a cost-effective method for pre-infection pathogen detection is required[2]. Such a detection concept must, on the one hand, offer the possibility of capturing pathogens from air and, on the other hand, be able to read out possible exposure information from a distance with a high sensitivity. The latter challenge is addressed in this contribution.

An obvious approach to solve this challenge is the use of radio frequency identification-based sensors combined with energy-harvesting technology[3]. While this concept provides wireless communication with promising results, it has the fundamental disadvantage that, in addition to the actual sensor functionality, active circuitry for energy harvesting, communication, and signal conversion is required. This negatively affects the reading distance, size, and complexity of the sensor[4–6]. Therefore, a reduction in the degree of complexity is required, for instance through the use of purely passive sensors. Here, the sensing concept can be based on the

same physical principles as for the required wireless communication, that is, electromagnetic (EM) waves. Sensors based on EM waves exploit a change in the refractive index (RI) in their immediate environment to detect pathogens. They can be highly specific, for example, using the binding of a target pathogen with a surface-bound specific receptor, thereby forming a thin film on the sensor surface[7,8].

The concept of RI sensors is one of the most promising candidates for the future of fast and label-free diagnostics and has been developed with different approaches, as explained in the following. Metasurfaces consisting of periodic split-ring resonators, in combination with spectroscopic systems, have been well investigated over the last decades. Immobilized biomolecules (e.g., nucleic acids, proteins) and cells on split-ring resonators can be detected by reading a resonance frequency shift[9–11]. Surface plasmonic resonance sensors exploit the interaction on the interface of metal and dielectrics and can detect thin films of immobilized biomolecules with a high sensitivity[12,13]. Furthermore, maturing fabrication has resulted in a high interest in whispering-gallery-mode resonator sensors for surface sensing, as they can achieve extremely high field concentrations[14,15]. However, these approaches have at least one severe disadvantage regarding the envisioned application: (i) low quality factors (Q-factor) in the case of metasurfaces and surface plasmonic resonance sensors or (ii) low field concentration at the site of the analyte in the case of whispering-gallery-mode resonators. As a result,

[1]Chair of Communication Systems (NTS), Faculty of Engineering, University of Duisburg-Essen (UDE), Duisburg 47057, Germany. [2]Institute of Digital Signal Processing (DSV), Faculty of Engineering, University of Duisburg-Essen (UDE), Duisburg 47057, Germany. [3]Institute of Technology for Nanostructures (NST), Faculty of Engineering, University of Duisburg-Essen (UDE), Duisburg 47057, Germany. [4]Organic Functional Molecules, Organic Chemistry, University of Wuppertal, Wuppertal 42119, Germany. ✉e-mail: yixiong.zhao@uni-due.de; jan.balzer@uni-due.de

their figure of merit (FOM), which evaluates their overall sensing capabilities and is positively correlated with the Q-factor and field concentration (Supplementary Note 1), are low. Biosensors based on optical photonic crystal (PhC) resonators are an exception, as they have the advantages of design flexibility, ultra-high Q-factor, and miniaturized geometric size[16]. It has been experimentally demonstrated that optical PhCs can successfully detect a concentration of 0.334 pg mL$^{-1}$ pancreatic cancer biomarker and have the lowest limit of detection among the methods discussed. This sensitivity can fulfill the COVID-19 biosensor requirements adequately[17,18].

In addition to their advantages, all discussed passive concepts have the severe disadvantage that reading the sensor either requires a proximity coupling or only works at short distances (i.e., in the centimeter range) from the reader. This contradicts the envisioned purpose of preemptive pathogen detection in a distributed sensing scenario, e.g., for the containment of airborne diseases such as Covid-19, which requires larger readout distances. The readout distance is defined by the radar equation (Supplementary Note 2). Here, especially its proportionality to $\sqrt{\lambda}$ ($\lambda$ is the wavelength of EM waves at the specific frequency) is an important physical limitation for the communication distance[19]. In consequence large wavelengths result in a long communication range on the one hand, but on the other hand require large antennas and sensors. Further, pathogens are very small compared to such wavelengths, which reduces the general sensitivity of the detection approach. From our point of view, frequencies which allow a good compromise between the discussed key parameters can only be found in the range of 100 GHz to 1 THz. This has motivated the research of PhC biosensors in the terahertz (THz) frequency range, where PhCs have been studied for wireless communication and biosensing applications[20–22]. Further, in previous study we have been able to demonstrate, that a sub-THz PhC slot resonator, working as a surface sensor, has a very high FOM when compared to other THz sensors[23].

Therefore, by combining high sensitivity with wireless readability, THz PhC resonators with a high Q-factor show immense potential for wireless biosensor applications. To prove the feasibility of this concept, here, PhC resonators are designed for the sub-THz frequency range and are implemented using 3D printed Al$_2$O$_3$, which is a material with excellent properties, i.e., high RI and low losses in the targeted frequency range[24]. In this framework, the wireless reading capacity and sensing performance of our sensors are studied. Remarkably, they can be read at up to 0.9 m and a 90° incident angle. Furthermore, we demonstrate that biomolecules can be wirelessly detected, which is a clear proof of principle for future wireless pathogen detection.

## Results

### Photonic crystals for remote sensing

In the following, the proposed method for highly sensitive remote detection of biomolecules is presented in brief. To achieve remote detection capabilities, wireless sensors can be distributed as landmarks to monitor the environment as shown in Fig. 1a. A hand-held or compact stationary system sends a querying signal and the sensor backscatters the signal containing information about its surroundings. Here, the sensor is a sub-THz PhC resonator with a specific resonant frequency $f_r$. When an analyte interacts with the sensor, the surrounding RI is changed by $\Delta n$. This RI change results in a change in the resonant frequency $\Delta f_r$ and allows the reader to identify and quantify a potential hazard. A high specificity for a certain pathogen (e.g. a virus) can be realized by immobilizing the specific bioreceptors on the sensor to capture the target molecule[17]. Figure 1b shows the proposed wireless sensor consisting of a PhC slot resonator for sensing, a PhC waveguide for energy coupling, and a dielectric rod antenna (DRA) for wireless communication. The sensor is based on a dielectric slab with periodic holes, which leads to a bandgap in a specific frequency range, called the bandgap frequency. Due to the wave confinement, the waveguide and the resonator can be realized by filling the holes and hence breaking the symmetry of the lattice[25]. When the wave radiated from the reader hits the wireless sensor, it is received by the DRA and coupled into the the waveguide. When the propagating wave in the waveguide hits the periodic holes, a portion is reflected and radiated directly through the DRA. The other part is coupled into the slot resonator. While the waveguide allows broadband operation, the resonator only supports frequencies which interfere

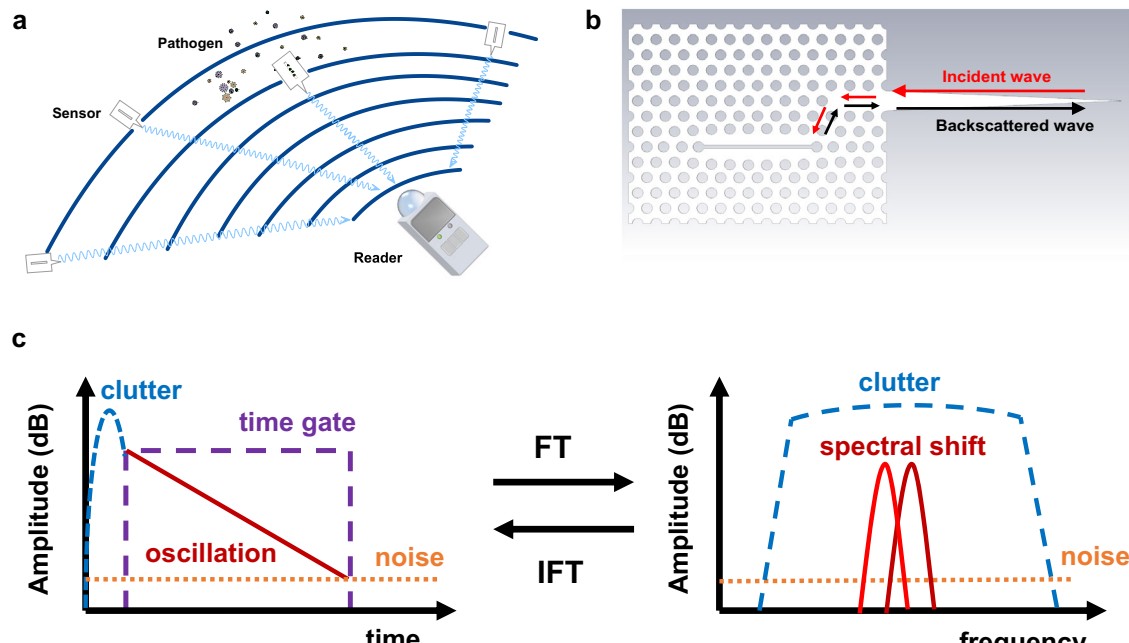

**Fig. 1 | Concept of the wireless sensor network. a** Schematic of the wireless sensors distributed in a room and interrogated by a reader. Airborne pathogens can be detected by the wireless sensors. **b** Layout of the proposed wireless sensor based on photonic crystal (PhC) slab consisting of a slot resonator, a waveguide, and a dielectric rod antenna (DRA). **c** Signal processing mechanism in time domain and frequency domain. Signals in time domain and frequency domain are transformed using Fourier transform (FT) and inverse Fourier transform (IFT). Note that the received signal at the reader includes the reflected clutter signal from the room, the sensor, the oscillating signal from the resonator, and the background noise. The clutter signal has a short duration, but a high amplitude overlapping the oscillating signal. It must be removed by a time gate to extract the oscillating signal from the sensor. The resonance spectral shift can be used to detect biomolecules, for instance.

**Fig. 2 | Simulation of the photonic crystal (PhC) resonator. a** Simulation of the resonator coupled with two PhC waveguides. The simulated electric field distribution is scaled in color. **b** The analyte under test (in red) is built on the wall of the slot. The adjacent holes around the slot are shifted to achieve a high quality factor (Q-factor). In our design, the first three adjacent holes along the resonator are shifted towards the outside with a displacement of $x_1 = 223\ \mu m$, $x_2 = 107\ \mu m$, and $x_3 = 107\ \mu m$, respectively. The holes below and above the resonator are shifted towards the outside with a displacement of $y_1 = 178\ \mu m$, $y_2 = 154\ \mu m$, and $y_3 = 130\ \mu m$, respectively. The optimized slot width is $w = 244.5\ \mu m$. **c** The simulated transmission parameter S21 and reflection parameter S11 of the resonators with the hole radius r = 0.30$p$ and r = 0.27$p$ ($p$ is the lattice period), respectively.

constructively, which leads to a resonance. A small portion of the energy is coupled back from the resonator into the waveguide and then radiated into the air by the DRA. If the energy loss of the dielectric material is exceedingly small, the resonator has a high Q-factor which is manifested by a long lasting oscillation in the time domain that is received by the reader (see Fig. 1c). The waves reflected from walls, obstacles and the sensor itself, on the other hand, have only a short duration since their energy has not been stored in the resonator. To remove the reflected interference i.e. clutter signal and extract the resonance information, the oscillating signal is time-gated and transformed to the frequency domain to read the resonant frequency, as shown in Fig. 1c. Possible adsorption of pathogens – ideally in a specific manner – in the resonator slot will result in a resonance shift.

### Resonator design

The resonator is designed and optimized using the EM simulation tool CST. To simplify the optimization and analysis, the resonator is coupled with two waveguides in the simulation and the transmission power is observed as shown in Fig. 2a. As mentioned earlier, pathogens such as bacteria and viruses can be specifically trapped and form a thin film on the surface of the sensor. To this aim, the sensor must be sensitive to the applied analyte layer. The sensing capability of the surface can be evaluated using an FOM, which is defined as

$$\mathrm{FOM} = \Delta f_r / (\mathrm{FWHM} \cdot \Delta n \cdot h_a) \qquad (1)$$

where FWHM is the full width at half maximum of the resonance in the frequency domain, $h_a$ is the thickness of an equivalent thin-film related to the captured analyte, and $\Delta n$ is the difference in RI of the analyte to air. The FOM can be improved by increasing electric field concentration at the site of the analyte and by increasing the Q-factor[23]. To enhance the field strength within the thin-film analyte, a slot is introduced in the middle of the resonator as shown in Fig. 2a, b. It can be seen that the maximum field strength at the resonant frequency is concentrated in the slot. For the design optimization, the thin-film analyte is deposited on the wall of the slot for enhancing the interaction as indicated by the red line in Fig. 2b.

To increase the field strength itself, the Q-factor has to be increased. The Q-factor of the sensor depends on material losses, radiation losses, and coupling losses due to the waveguides[25]. To realize an efficient PhC, a high value of $\varepsilon_r$ and a low loss tangent are desired. Here, we use alumina with $\varepsilon_r = 9$ and loss tangent = 0.00022 which has the further benefit that it can be 3D printed with a resolution of 25 μm using lithography-based ceramic manufacturing (LCM), a fabrication technique for fast prototyping[26]. To prove our proposal, a PhC resonator with a lattice period $p = 1100\ \mu m$ and a thickness $t = 0.5 p$ is chosen for W band (75–110 GHz) operation. The radiation losses can be minimized when a smooth Gaussian distribution of the electric field is achieved along the resonator[27]. This is realized by a resonator length of 6 periods. To further improve the Q-factor, the

neighboring holes are shifted around the slot as shown in Fig. 2b. The displacement of the holes and the width of the slot are optimized for maximum Q-factor by sweeping the structural dimensions. Further details of design processes can be found in our previous work[23]. In addition, the coupling losses depend on the number of hole rows between the waveguide and the resonator, the diameter of the holes, and the number of waveguides used for the coupling. The more rows of holes and the larger the diameter, the better the field confinement within and the lower the coupling to the resonator. As a result, the Q-factor increases. However, this weakens the transmitted signal. Since the weakened transmitted signal leads to a reduction of the detected signal at the reader and hence reduces the reading distance, a large number of rows of holes for higher Q factors is not considered in this work. Here, 3 rows of holes are chosen and different hole radii $r = 0.30 p$ and $r = 0.27 p$ are compared. The simulated transmission parameter S21 is shown in Fig. 2c. The resonant frequencies are 97.440 GHz and 95.144 GHz, with a Q-factor of 2830 and 1740, respectively. As expected, the resonator with larger holes has a higher Q-factor with a lower peak magnitude. This indicates that the Q-factor should be limited to achieve a reasonable peak magnitude. It should be noted that the reflection parameter S11 does not explicitly shows the resonant frequency $f_r$. This is because the reflected signal at the input of the PhC waveguide is superimposed with the signal from the resonator. However, $f_r$ can be easily read in the transmission parameter S21.

To study the sensitivity of the two resonators, a thin film of analyte with varying RI $n$ and varying thickness $h_a$ is applied on the wall of the resonator. Since the sensor is much thicker than the analyte, the analyte is set as an independent mesh in the CST simulation. First, the resonator is simulated with $h_a = 0.5\ \mu m$ and $n$ is varied from 1 to 2, as shown in Fig. 3a. Since the analyte layer is very thin compared to the wavelength, the resonance shifts are assumed to be reciprocally related to the RI of the analyte[23]. Next, $h_a$ is swept from 0.1 to 1 μm while $n$ is held constant ($n = 1.8$) to investigate the sensitivity to the analyte as the thickness varies, as shown in Fig. 3b. It can be seen that the resonance shifts are linearly related to thickness. Due to the higher Q-factor, the resonator with $r = 0.30 p$ shows a higher resonance shift and FOM (1.69 RIU$^{-1}$ μm$^{-1}$) than the resonator with $r = 0.27 p$ (FOM = 0.94 RIU$^{-1}$ μm$^{-1}$).

### Remote reading design

A long reading range and a large reading angle will allow high flexibility and reliability in pathogen detection. According to the radar equation, the maximum reading range increases with enhanced radar cross section of the sensor (Supplementary Note 2). For a long reading range, the radar cross section of the sensor can be increased by increasing the taper length of the DRA. Here, a high-gain DRA is used as shown in Fig. 4a. Its wider end is connected to the PhC waveguide and has the same cross section as the waveguide (1.25 mm width and 0.55 mm height). The other end terminates in a tip that provides a good transition from the PhC waveguide to the air.

Increasing the length provides a smoother transition, but complicates fabrication, enlarges the sensor, and reduces the readable angular range. Here, lengths of 7, 14, and 21 mm are chosen for prototyping. The far-field radiation from the DRA excited by a PhC waveguide is simulated as shown in Fig. 4a. The DRA shows a radiation pattern with a main lobe in the direction of the taper tip with antenna gains of 9.3, 11.5, and 13.0 dBi, respectively. However, the DRA is excited by the resonator and the excited waves have a different EM mode as the fundamental mode of the PhC waveguide, resulting in a mismatch of EM field distribution and a dispersed radiation pattern. To study the far-field radiation, the resonator is simulated with one DRA for excitation which is inserted into the excitation port and another DRA for radiation, as shown in Fig. 4b. Its radiation pattern is divided into several lobes. As a result, the antenna gain is lowered. The far-field gain in the $\varphi$ and $\theta$ planes (see Fig. 4b) is plotted in Fig. 4c and d, respectively. The radiation gain is reduced to −2.7, 5.1, and 4.5 dBi in the

direction of the DRA tip, respectively. As the length of the DRA increases, the radiation pattern tends to be a main lobe similar to that excited by a PhC waveguide. Furthermore, the radiation covers a large angular range.

## 3D printing of ceramic

To enable rapid prototyping and have maximum freedom in the design, a 3D printing process is used to fabricate the samples. Compared with micro- and nanofabrication, the 3D printing process has lower cost and a shorter fabrication time for a small volume of samples. The samples are fabricated from alumina ($Al_2O_3$) due to its high permittivity and low loss. Details on the fabrication are described in Methods.

Figure 5a shows the fabricated sensors S027T7, S027T14, S027T21, S030T7, S030T14, and S030T21 (S030 and S027 denote the hole radius of $r = 0.30\,p$ and $r = 0.27\,p$, respectively; T7, T14, and T21 denote the DRA length of 7, 14 and 21 mm, respectively). The body of the sensor excluding

**Fig. 3 | Simulated resonance shifts with the hole radius r=0.30p and r=0.27p (p is the lattice period), respectively. a** Resonance shifts and their reciprocal fitted curves with varying refractive index (RI) of the analyte. **b** Resonance shifts and their linear fitted lines with varying thickness of the analyte.

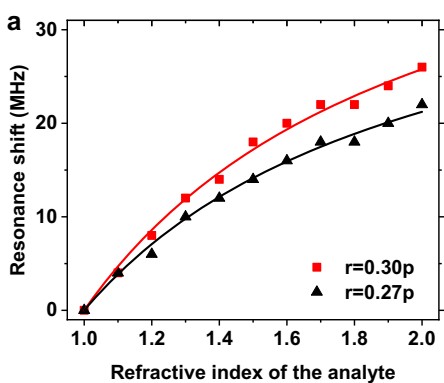
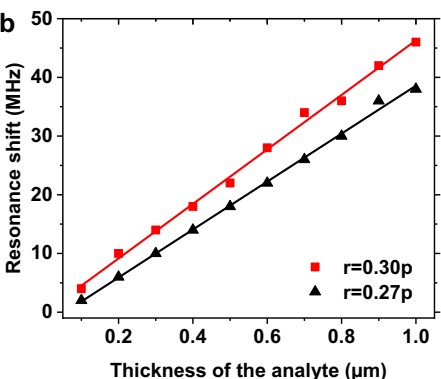

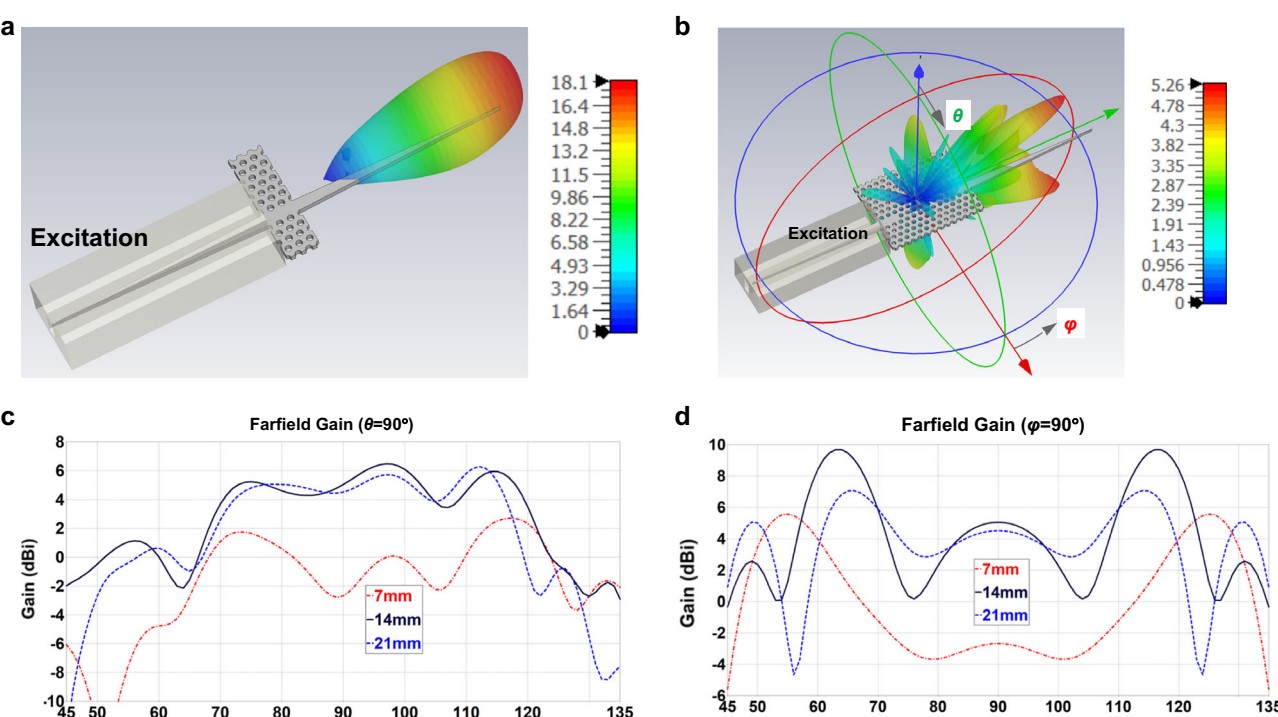

**Fig. 4 | Simulated radiation gain. a** The radiation pattern of the dielectric rod antenna (DRA) excited only by a photonic crystal (PhC) waveguide. **b** The radiation pattern of the resonator with two tapers with changing angle $\varphi$ and $\theta$. The resonator is excited through the taper inserted into a waveguide and radiates waves through the

DRA on the right side. **c, d** Simulated far-field gain of the resonator of different taper lengths (7, 14, and 21 mm) with varying angle $\varphi$ and with varying angle $\theta$, respectively.

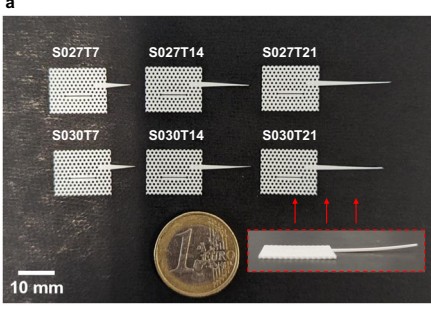
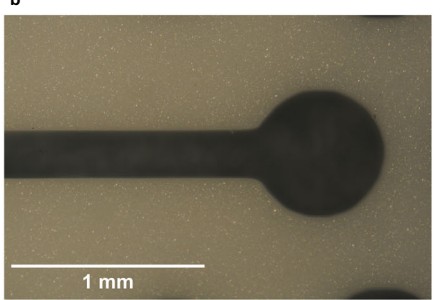
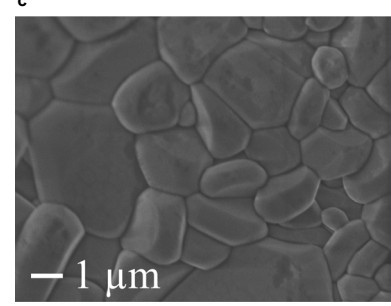

**Fig. 5 | Fabricated samples. a** Photo of the manufactured samples with a one euro coin to show the proportions (S030 and S027 denote the hole radius of $r = 0.30\,p$ and $r = 0.27\,p$, respectively; T7, T14, and T21 denote the DRA length of 7, 14 and 21 mm, respectively). The photo in the red box shows the side view of Sample S027T21. **b** An example optical microscope image of a fabricated slot of Sample S030T14. **c** SEM image of a fabricated sample.

the DRA is only about 12 mm × 15 mm. It can be seen that the 21 mm DRA is slightly warped upward. This phenomenon occurs only for samples with long DRA (i.e., when the ratio between length and width of a feature is large) and is related to the temperature gradient in the sintering furnace. Figure 5b is showing an image of the slot taken by an optical microscope. The width of the slot and the hole radius are measured to be about 245.8 μm and 330.1 μm, respectively, showing only a slight deviation from the design. Furthermore, as seen from the scanning electron microscope image in Fig. 5c, the fabricated samples have almost no open porosity indicating high quality alumina.

### Wireless reading capacity

A long wireless reading range is an important criterion for use as a remote sensor. To study the wireless reading capacity, a vector network analyzer (VNA) connected to a horn antenna is used as the reader. The data collected by the VNA in the frequency domain is transformed to the time domain by inverse Fourier transform. The different wireless sensors are placed at a distance $L$ between the tip of the DRA and the edge of the horn antenna and aligned to the center of the horn antenna by using a laser pointer. Further details of the measurement setup are given in Methods, Supplementary Note 3, and Supplementary Fig. S1. Figure 6a shows an exemplary measurement result of S027T14 in the time domain at a distance of $L = 0.5$m. The first signal cluster results from the mismatch of the horn antenna to the waveguide of the VNA. The second signal cluster is due to the reflection of the DRA and the termination of the PhC waveguide and is hence linearly related to the distance $L$. It is followed by the decaying oscillatory signal from the resonator. To reconstruct the oscillatory signal, the time-domain signal is filtered with a Tukey window starting 1 ns after the second cluster starts and ends before the third cluster, as shown in Fig. 6a. The start time of the third cluster is about 20.6 ns and does not change when the distance $L$ is changed. This is due to an internal reflection of the measurement system. To intuitively observe the spectral trend in the time-domain, the time-gated signal is processed using the short-time Fourier transform. Figure 6b shows the change of its local frequency content over time. The resonance peak has a high magnitude compared to background noise and can be easily observed. As expected, the magnitude of the resonance peak decreases exponentially over the time. Without filtering, this oscillating signal is overlapped by the reflection clusters in frequency domain and cannot be identified from the raw reflection parameter S11 as shown in blue in Fig. 6c. After time gating, the reflection clusters are eliminated and the resonance peak with a higher magnitude than the clutter can be easily observed as shown in orange in Fig. 6c. Its maximum magnitude and resonant frequency provide information about the sensor's immediate environment and enable pathogen detection, for example.

The fabricated sensors with different taper lengths and hole radii are implemented to investigate the influence of the reading distance $L$, which was varied from 0 m to 0.9 m with steps of 0.1 m. For comparison, the duration of the time gate is kept constant at 10 ns. To observe the strength of the backscattered signal, the maximal magnitudes at resonant frequency are calculated and plotted in Fig. 6d. As expected from the radar equation, increasing $L$ weakens the received signal at the reader. All 6 samples are readable up to a distance of 0.9 m. A larger reading distance is limited by the instruments noise floor, output power, and antenna gain. It can be seen that a longer taper leads to a higher magnitude of the resonance peak. Comparing the sensors with $r = 0.30\,p$ and $r = 0.27\,p$, it can be seen that the sensors with the smaller holes show a higher magnitude due to the stronger coupling between the resonator and the waveguide. Thus, if the time gate has a short duration, the sensor with the lower Q-factor will have a higher backscattered energy.

Next, the effect of the duration of the time gate is investigated by using a constant start time and a varying end time. For this purpose, the samples are positioned at a distance of 0.5 m from the reader. The time window starts at 5 ns. The length of the time window is increased from 10 ns to 20 ns with an increment of 1 ns. Figure 6e shows that the peak value of S027T14 stops increasing from 17 ns, while the peak value of S030T14 increases and is higher than that of S027T14 at 20 ns. Looking at the trend of the measured Q-factor in Fig. 6f, the Q-factors of both resonators remain identical up to 15 ns. When the time gate is longer than 15 ns, the Q-factor of S030T14 increases, while the Q-factor of S027T14 decreases. The reason is that the resonator with smaller holes has a smaller characteristic Q-factor and its oscillation signal decreases faster in the time domain. The noise is no longer negligible after 15 ns and interferes destructively with the signal of the resonator. As a result, the measured Q-factor is reduced. Thus, as the duration of the time window increases, the sensor with the higher Q-factor has a higher magnitude and a higher measured Q-factor.

In addition to the backscattered energy, a large reading angle plays an important role. A large reading angle ensures a reliable reading of the sensor even under non-ideal incidence angles and thus increases flexibility. To investigate the dependence on the angle of incidence, the resonator is placed at a distance $L = 0.3$m from the reader and is rotated from −45 to +45° with increments of 2.5°. At 0°, the DRA is aligned towards the reader antenna. The resonator rotated with varying $\varphi$, as shown in Fig. 7a. Comparing sensors with varying rod lengths reveals that a longer rod gives a higher magnitude but a smaller beam width in the $\varphi$ plane. Next, the axis of rotation is changed to be parallel to the PhC. The dependence on the $\theta$ angle is plotted in Fig. 7b. Similar to the observations in the $\varphi$ plane, the longer rod in the $\theta$ plane shows a higher amplitude and a smaller beam width. The measured magnitude has a slightly different shape because the DRA was slightly warped during fabrication. However, the measurement results with varying $\varphi$ and $\theta$ show that the wireless sensor can be read over a wide angle of incidence. In the measurement setup, a line-of-sight between the sensor and the antenna is used. Obstacles within the line-of-sight may attenuate the signal severely and affect the detection. However, the large reading range and acceptance angle of the sensor allow for a solution to this problem by moving the reader to read every sensor distributed in the monitoring area. Additionally, not every sensor needs to be read at all times.

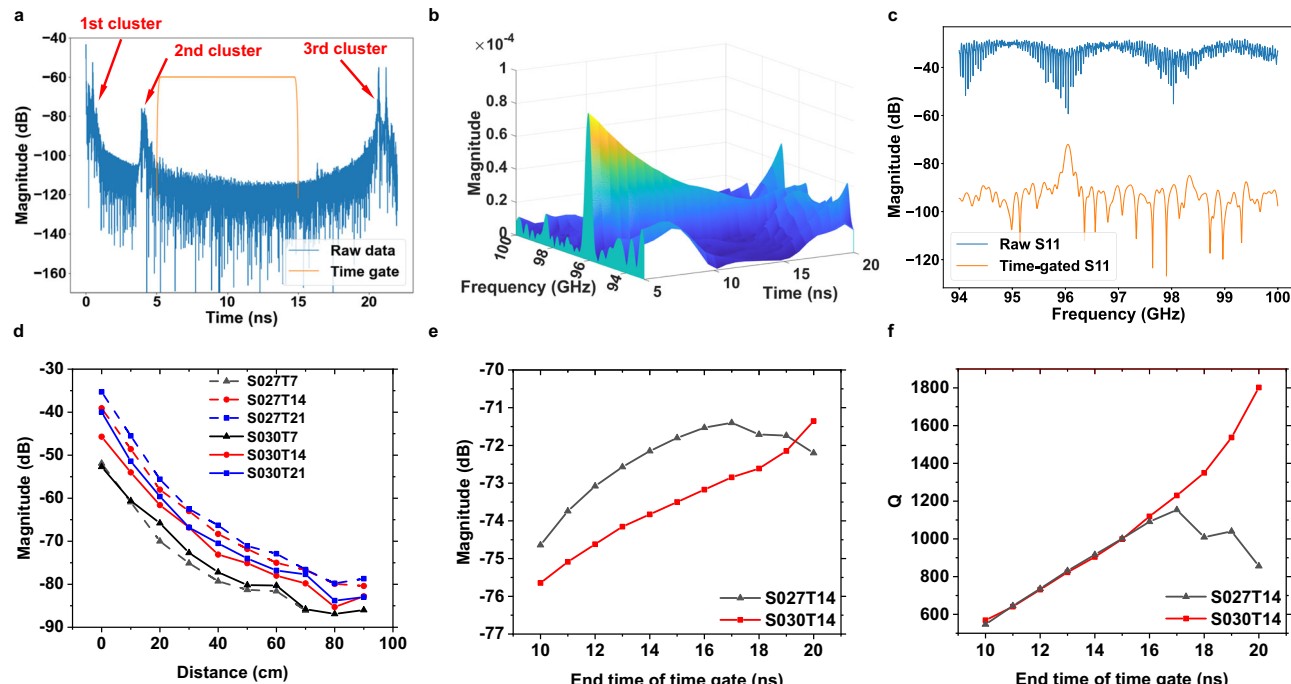

**Fig. 6 | Data processing of the measured reflection parameter. a** Measured reflection of S027T14 in time-domain (blue) and the time gate used (orange). The time gate starts from 1 ns after the second cluster and ends before the third cluster. **b** Spectrogram of the time-gated signal of S027T14 depicting the spectral trend in the time domain. **c** Raw reflection parameter (blue) and time-gated reflection parameter of S027T14 (orange) in the frequency domain. **d** Measured resonance magnitudes of all fabricated samples with varying distance. **e** Measured magnitude with increasing end time of the time gate. **f** Measured quality factors (Q-factor) with increasing end time of the time window.

---

**Fig. 7 | Measured magnitudes and their polynomial fitted curves with different taper lengths.** **a** Magnitude with varying angle $\varphi$; **b** Magnitude with varying angle $\theta$.

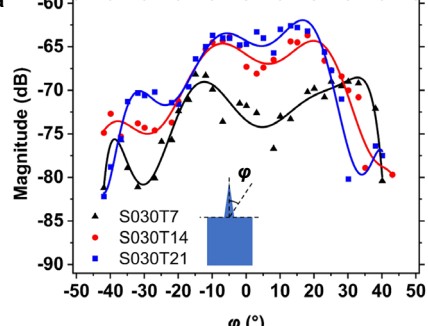
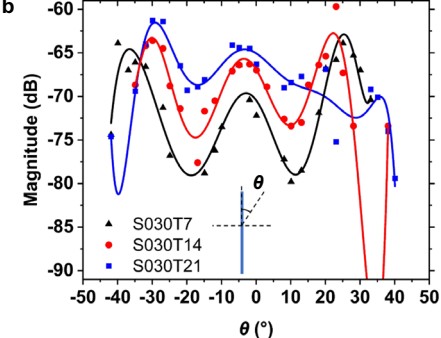

---

## Wireless biomolecule sensing

To characterize the wireless sensing capacity, the sensors are placed at 0.5 m distance to the horn antenna and loaded with varying amounts of bovine serum albumin (BSA), a model protein classically employed for proof-of-principle experiments. For the measurement, 1.5 μL of the BSA solution are pipetted into the slot of the sensor. After drying a protein layer remains on the slot wall and leads to a resonance shift. Protein solution concentrations of 2.4, 4.8, 7.2, and 9.6 g L$^{-1}$ are prepared to reach various biomolecular film thicknesses after drying. The thickness of the deposited BSA $h_a$ is estimated to be 0.5, 1.0, 1.5, and 2.0 μm, respectively. A detailed description of the measurement process is given in Methods. The resonance shifts due to the deposited BSA are plotted in Fig. 8. It can be observed that the resonance shifts of S030T14 (red) increases with increasing BSA concentration. The mean value of the resonance shifts can be linearly fitted with a reasonable coefficient of determination (0.9878). The slope of the fitted line is 5.2 MHz μL μg$^{-1}$. Considering the estimated thickness $h_a$, the sensitivity is 25 MHz μm$^{-1}$. Since the frequency resolution of the measurement device is 1 MHz, a 40 nm thick film can be detected. However, the measurement

shows tolerances coming from the manual liquid transfer, such as the transferred volume, the deviated position and non-uniform distribution of the dried protein, and system noise. A 2.1 MHz standard deviation at 0 μg μL$^{-1}$ concentration can be related to the uncertainty of the complete measurement setup. It can be noted that the standard deviation increases with increasing concentration. The reason is that increasing concentration leads to the saturation of the protein in the water and more suspended combined protein (Supplementary Note 4 and Supplementary Fig. S2). If the resonance shift must be higher than 3 times the standard deviation to achieve a 99.73% possibility of correlation, the resulting acceptable confidence is 1.8 μg or 250 nm thick biomolecule film. An essential part of the measurement tolerance relates to variations in ambient temperature and device noise. Temperature fluctuations can lead to resonant shifts. As presented in our previous work[23], the influence of the environment can be minimized by introducing a reference resonator. The difference between the resonant frequencies of the reference resonator and the resonator for sensing remains almost constant with varying temperature. This approach can effectively minimize the standard deviation enabling a decrease in the

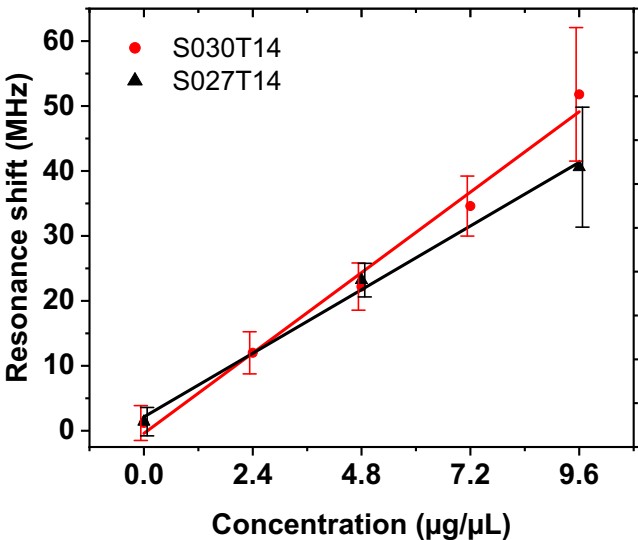

**Fig. 8 | Statistic plot (displays the mean as a dot and standard deviation) and linear fitted line of the measured resonance shift.** Measured resonance shifts change with varying concentration of the bovine serum albumin (BSA) solutions deposited on the sensors S030T14 and S027T14. The number of independent experiments is $n = 5$. The measured results are linearly fitted.

minimum detectable layer thickness. Furthermore, the sensitivity can be further improved by increasing resonant frequency [28]. On the other hand, S027T14 (cf. Fig. 8, black curve) shows a slightly lower resonance shift than S030T14, as expected from the simulation results. As a summary, the sensor with a higher Q-factor shows an advantage in both sensitivity and readability.

## Discussion

In this work, a PhC-based sensor was presented that combines high sensitivity to thin biomolecule films with long wireless read distance. In this approach, EM waves are used for both sensing and communication. This simplifies the design and enables passive sensors with maximum compactness. The proposed PhC slot resonator shows a high Q-factor which leads to a higher FOM compared to other THz sensors [10]. The high Q-factor results in a long oscillation time, which helps to remove the backscattered clutter of the environment. By using time gating, the oscillating signal can be easily extracted from the background signal without room calibration. To prove this concept, PhC sensors with different dimensions were fabricated using LCM 3D printing. The measurement results show a long reading range, a large reading angle, and good wireless sensitivity for a model BSA protein. This enables the distribution of remote sensors in a large area, i.e. a room or hallway, for monitoring airborne pathogens. In comparison to sensors connected to a central reader with cables or those requiring reading in proximity, the proposed remote sensor offers easy and efficient reading capabilities, making it highly suitable for large-scale deployments. In addition, the proposed sensor can operate in critical environments without an active chip and battery and has a long lifetime. The passive remote sensor is non-electronic, compact, and inexpensive, making it easy to dispose of. This gives it an advantage over solutions that use active circuits. Since there are currently no other methods to realize non-electronic wireless sensors, this concept will play a key role in distributed wireless sensing applications.

Our future effort will be dedicated to improving the performance of the proposed sensor and considering its practical applicability. Firstly, the detected signal can be enhanced by employing a high-gain antenna and optimizing the coupling efficiency between the DRA and the PhC waveguide. For instance, a planar dielectric antenna designed using effective medium theory and fabricated on the same PhC slab can achieve a gain of up to 20 dBi [29]. This approach will enhance the gain of the DRA and hence the maximum detection range. Furthermore, the coupling efficiency between

the DRA and the PhC waveguide can be improved through a mode transition, such as a tapered PhC waveguide. Secondly, an algorithm can be developed aimed at improving the detection reliability on a low-cost, small single-board computer on the reader side. For example, a calibration-free detection technique based on short-time Fourier transform was proposed for chipless radio frequency identification tags to improve their detection reliability [30]. These methods will extend the maximum reading range. Furthermore, in future work, the sensitivity of the current design should be improved to achieve a reliable detection of a monolayer of biomolecules and proteins. To achieve this, its resonant frequency can be increased and the analyte can be deposited on a larger area of the sensor, including the walls of all the holes and the top/bottom surface of the resonator. This will also facilitate the immobilization process in the next steps. Detailed studies on the effects of the resonant frequency and the covered area can be found in Supplementary Note 5 and 6 and Supplementary Figs. S3–9. Considering the future potential, the sensor can be manufactured in large quantities on high resistance silicon wafers using a standard process such as deep reactive ion etching (DRIE) to reduce costs and increase throughput. Using DRIE, it is possible to achieve an aspect ratio of 30 or more, while the proposed sensor only has an aspect ratio of 2.2. This method can be easily transferred to PhC resonators. In preliminary investigations, we characterized 9 DRIE-etched 250-GHz PhC resonators randomly selected from the same wafer. The standard deviation of their resonant frequencies was only 0.1% of the average resonant frequency. This result was obtained without optimizing the DRIE process parameters. The throughput per wafer can be further increased by increasing the resonant frequency as this reduces the size of the sensor. With the advantage of low cost fabrication, the wireless sensor can be disposed once the pathogen is captured and detected. This eliminates the need for complex cleaning processes for reuse. Furthermore, the proposed concept can be extended with a simplified radar system serving as the reader and multiple sensors tuned to different resonant frequencies for multiplexing their signals and identify their locations [31–33].

## Methods
### Simulation setup

A frequency domain solver in CST is chosen, because it can get rapid and precise results of the resonant frequency compared to a time-domain solver. Furthermore, using a frequency domain solver, the resonator and the thin-film analyte can be meshed independently using tetrahedral mesh type. The two tapers or DRAs of the resonator are inserted into two WR10 waveguides for W band frequency (75–110 GHz), whose dimensions are 1.27 mm in height and 2.54 mm in width (Supplementary Note 7 and Supplementary Fig. S10).

### Fabrication

A photosensitive LithaLox360 slurry and a CeraFab 7500 printer from Lithoz are utilized. The printer is an LCM 3D printer with a resolution of 25 μm and therefore is suitable for rapid prototyping of the delicate structures required for the sensor. After a controlled layer-by-layer polymerization of the slurry, the obtained green bodies are cleaned, slowly heated up to 1600 °C and held for 2 hours to fabricate dense alumina samples. More details about these steps can be found in our previous work [34].

### Measurement setup

The resonator is measured using a VNA ZVA67 and a frequency conversion module ZC110 extending to the W band (75–110 GHz). A 26 dB horn antenna is mounted on the extender for wireless measurements. The frequency resolution is set to 1 MHz. The wireless sensor is mounted on a 3D printed holder made of acrylonitrile butadiene styrene. The effect of the support is minimal because only 0.5 mm of the sensor is inserted into the mount and the signal at the edge is very weak after reflection through several holes of the PhC forming a frequency band gap. The taper of the sensor is aligned to the center of the horn antenna using a laser as shown in Supplementary Fig. S1. A distance $L$ is kept between the tip of the taper and the edge of the horn antenna.

## Signal processing

The VNA measures the reflection parameter in frequency domain. It is transformed to the time domain using the inverse fast Fourier transform. To remove the clutter, a Tukey filter with a coefficient of 0.05 is created as a time gate. It has a flat amplitude in the center and cosine form at both ends to reduce the ripple in the frequency domain. The measured signal is multiplied with the time gate in the time-domain and then transformed back to the frequency domain.

## Biomolecules

The used BSA is bought from Carl Roth. It has a purity of higher than 98% and a molecular weight of 66.5 kDa. BSA is carefully dissolved in deionized water by using a shaker for 2 h, to avoid coagulation.

## Sensing measurement steps

First, the sensor is aligned to the center of the horn antenna. Using the function of the time window on the VNA, the resonance peak can be observed. The raw data of the reflection parameter S11 is saved. Then, BSA solution is pipetted into the slot of the resonator. In this step, the solution is slowly dropped into the two holes connected to the slot and the liquid spreads into the slot. After optimization, 1.5 μL is found to be a proper volume that fills the slot well and leaks only minimal. The fabricated samples have almost no open porosities. Hence, it is correct to assume that during the experiment the liquid will stay in the slot and cannot penetrate into alumina. After dropping, the resonance peak is depressed due to the loss of water. As the solution dries up, the amplitude of the resonance peak increases. The analyte is completely dried when the resonance peak remains constant. The layer thickness was approximated by using $h_a = V_{drop} \cdot C/\rho/S$, where $V_{drop}$ is the volume of one drop, $C$ is the BSA solution concentration, $\rho$ is the density of dried BSA, and $S$ is the area of the wall of the slot. Then, the raw S11 of the loaded resonator is read from VNA. Finally, the sensor is taken away from the support and washed to remove the deposited protein. After this, the sensor can be reused for the next measurement. The measurement for each concentration is repeated 5 times to assess the reliability of the measurements.

## Statistics

During the sensing measurement, standard deviation was used to evaluate the tolerance of the measurement setup. The measurements were repeated 5 times.

## Data availability

The data that support the findings of this study are available from the corresponding author Yixiong Zhao upon reasonable request.

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

## Acknowledgements

This work was supported in part by the Deutsche Forschungsgemeinschaft through TRR 196 MARIE Projects M01, M05, and C09, under Grant Project-ID 287022738, in part by the Ministry of Culture and Science of the State of North Rhine-Westphalia (MKW NRW) through Project terahertz.NRW, and in part by the Open Access Publication Fund of the University of Duisburg-Essen. The project terahertz.NRW was supported by the program Netzwerke 2021, an initiative of the Ministry of Culture and Science of the State of Northrhine Westphalia.

## Author contributions

T.K., J.C.B. and Y.Z. conceived the concept. Y.Z. designed and simulated the wireless sensor. M.S. fabricated the sensor. G.B. prepared the analyte under test. Y.Z. and A.A.A. designed the measurement setup. Y.Z. performed the experiments. Y.Z. and J.C.B. analyzed the data. Y.Z. wrote the manuscript with contributions from all authors. T.K., J.C.B., N.B. and G.D. contributed to funding acquisition and supervision.

## Funding

## Competing interests

The authors declare no competing interests.
