## [Peer Review File · Communications Engineering]

Reviewers' comments:

Reviewer #1 (Remarks to the Author):

The authors present a beautiful work on a wireless photonic crystal-based sensor using sub-THz radiation. The scheme of the experiment is good and the proposal is ambitious. The experiment was carefully designed by designing the high Q factor cavity with near near-perfect reflection background to have a maximum S11 read-out signal. I have some comments for the authors to dwell on and respond to.

1. A weakness that I see in this otherwise bold experiment is the read-out THz signal power which is really at the limit of high-frequency VNA (-90 dB) and is very close to the noise floor. Is it possible for the authors to improve the signal, for example, the data in Fig. 6d and Fig. 7? The signal in 6d is better for smaller distances but the data at 0.5 m looks weak and noisy, although the time gating strategy is a good one.

2. Will a higher Q PC cavity improve the sensitivity further of this wireless biosensor? Is the Q factor in this design well-optimized? What are the limitations in getting higher Q in these PCs?

3. The authors have missed some recent developments in THz silicon photonic crystals for sensing/ biosensing applications that could be appropriately referenced: APL 121, 011101 (2022); APL 123, 033705 (2023).

Reviewer #2 (Remarks to the Author):

Manuscript: COMMS-23-0344A

Title: 3D Printed Sub-THz Photonic Crystal for Wireless Passive Biosensing

Authors: Y. Zhao et al.

Summary: The authors demonstrate a sub-THz photonic crystal resonator to remotely sense proteins or other bio-molecules. The sensor has a reading range of max 0.9 m and an elevation and azimuth acceptance angle of nearly 90 degrees. It is supposed to be non-electronic, compact, and low-cost.

Comments: The paper reports on an interesting concept for remote sensing of bio-molecules. While the general concept is interesting and original, there are several comments the authors must address prior to publication.

In the abstract the authors claim: "... This will enrich the functionality of the Internet of Things and may help to mitigate pandemics in the future..." I find both claims far-fetched and unsubstantiated. I also don't understand what this has to do with Covid. The only effect the authors have demonstrated is that the resonance peak of a resonator shifts when the index of refraction of the material inside the resonator/slot changes. A well-known effect, nothing new!

What is original is that the resonance shift, via a coupler that connects free-space to the resonator, can be detected remotely up to a distance of nearly one meter.

The observed resonance shift, per se, is absolutely unspecific and reveals no information whatsoever on the sample other than the magnitude of the change in index of refraction. Whether this is caused by water, salt, sugar or some protein can not be inferred from this measurement. At some point the authors claim that specificity comes into the game when the surface is functionalized so that only very specific bio-molecules can bind to it. This would result in a more or less mono-molecular layer of bio-molecules or proteins etc. and the setup - as presented - is not sensitive enough to measure such small modifications to the index of refraction. Recall the minimum layer thickness required to observe a detectable resonance shift is 250 nm. Hence, claim and reality do not match and require further clarification. I would find it mandatory to repeat the measurement with a functionalized surface/slot and to substantiate the claims with a realistic measurement. Otherwise the claims have to be reformulated reflecting the realities of the presented results.

What would happen if the voids of the photonic crystal also fill with the same pathogen? A shift in band gap? What would be the consequence? Have the authors tested - via simulation - what happens if the bio-molecules not only fill the slot but also the voids between the pillars of the photonic crystal structure?

Moreover, the remote sensing aspect requires clarification. Suppose the device is functionalized for detection of some specific pathogen and at some point in time will be fully covered, i.e. saturated. Then the device becomes useless unless cleaned. This clearly limits the use of the detector to some very specific applications, which the authors should be elaborate on. Is the remote sensing aspect really that important then? Time stamp information is directly linked to the read-out frequency.

Further, I would expect some more details on the simulations and the material parameters used.

Reviewer #3 (Remarks to the Author):

The authors propose and demonstrate an interesting RI biosensing concept with a long stand-off distance.

The device uses a PhC designed to operate in the W band (75-110GHz) regime.

The authors also 3D fabricate a structure and demonstrate a proof-of-concept device.

While interesting, this work seems better suited to a biosensors or a devices journal as the novelty in the communication strategy and its fit for this journal is not clear.

If the authors choose to modify and resubmit the manuscript either here or to another journal, it would benefit the authors to clarify the following observations to further strengthen their manuscript:

(a) the sensor assumes a clear line-of-sight between the sensor and the detector. However, in real-world applications, there are obstructions (including human subjects and walls) between the sensor and the detector, specifically for the proposed application. The presence of such obstacles has shown to severely attenuate W-band signals. How can such a device be implemented, in say, crowded areas? Why would it be disadvantageous to detect a signal in close proximity and wirelessly transmit the electrical signal given IoT promises to make devices cheap and connected?

(b) as opposed to structures such as RFIDs that can be produced in the millions in a short amount of time at a very-low cost, the proposed 3D manufacturing approach will only be able to fabricate a low volume of devices irrespective of the claim made by the authors in terms of production speed (taking several hours for curing itself). The 3D manufacturing approach just cannot match screen-printing, for example in terms of throughput.

(c) based on the above facts, the authors should comment on the potential deployability of millions of devices and cost of manufacturing.

(d) the authors point out energy harvesting and signal processing as some of the disadvantages for the current state-of-the-art electronic sensors. One can also argue that deploying a network analyzer or a detector in the W-band comes with significant investments and operation know-how, plus the added difficulty of aligning the detector and the sensor structure with high precision (that is one detector per a few sensors, which is not practical). Also, how would this system multiplex signals from multiple sensors without deploying complex electronic processing equipment at the detector or the sensor head?

(e) PhC structures are very sensitive to any small changes in structural parameters or any changes in the environment. How does change in ambient temperature/humidity affect the performance of the devices?

(f) a stand-off distance of 0.5m is still not sufficient, especially given the transmissivity and diffusivity of viruses in aerosol form.

Summary: while the work tries to showcase a biosensing concept using a high-freq PhC structure and claims to achieve a large sensing distance, the complex fabrication process (slow speed, low repeatability, high cost) and practical constraints in deploying hundreds and thousands of such sensors make them less effective or attractive compared to the state-of-the-art devices.

Reviewer #1 (Remarks to the Author):

The authors present a beautiful work on a wireless photonic crystal-based sensor using sub-THz radiation. The scheme of the experiment is good and the proposal is ambitious. The experiment was carefully designed by designing the high Q factor cavity with near near-perfect reflection background to have a maximum S11 read-out signal. I have some comments for the authors to dwell on and respond to.

We thank the reviewer for reviewing our manuscript and providing suggestions to improve our work. According to the reviewer's comments, we have responded point-by-point and revised our manuscript as shown in the following. Additions to the text of the manuscript, individual words, and variables that have been replaced or added are highlighted in red.

1. A weakness that I see in this otherwise bold experiment is the read-out THz signal power which is really at the limit of high-frequency VNA (-90 dB) and is very close to the noise floor. Is it possible for the authors to improve the signal, for example, the data in Fig. 6d and Fig. 7? The signal in 6d is better for smaller distances but the data at 0.5 m looks weak and noisy, although the time gating strategy is a good one.

Answer: We thank the reviewer for pointing this out. In our manuscript, we have developed a proof-of-principle remote sensor prototype. Our current focus is on enhancing the detected signal to increase the reading distance. We would like to address this concern by discussing two potential approaches:

(a) The first method is that the received signal can be enhanced by increasing the gain of the dielectric rod antenna (DRA) and optimizing the coupling between the photonic crystal (PhC) waveguide and the DRA. In Fig. 6d, we observe that the sensor with a higher gain (about 5 dB) shows a higher magnitude (about -70 dB at 0.5 m), indicating that increasing antenna gain enhances the detected signal. According to the literature (doi: 10.1063/1.5060631), a planar dielectric antenna fabricated on the same photonic crystal slab can be designed using effective medium theory to reach a gain of 20 dBi. Hence, our current DRA has the potential for a 15 dB improvement. Additionally, the coupling between the PhC waveguide and the DRA can be improved through a mode transition, such as a tapered PhC waveguide.

(b) Secondly, we are studying a data-processing algorithm to enhance detection reliability. In Fig. 6c, the resonant peak for a 0.5 m distance is about 15 dB higher than the noise level. The resonant frequency can be identified by searching the maximum peak. For a long distance with a low signal-noise-ratio, the resonant signal can be enhanced and the noise can be reduced. In the literature (doi: 10.1109/TAP.2020.3016160), Lin *et al.* developed a method based on short-time Fourier transform (STFT), averaging, and filtering to detect chipless RFID tags. This approach can potentially be applied to improve the detection reliability in our proposed sensor.

In summary, the current reading distance achieved by our proposed sensor is longer than the state-of-the-art and can be further improved by optimizing the sensor's structure and developing suitable algorithms.

We have updated the Discussion of the manuscript by:

Our future effort will be dedicated to improving the performance of the proposed sensor and considering its practical applicability. Firstly, the detected signal can be enhanced by employing a high-gain antenna and optimizing the coupling efficiency between the dielectric rod antenna (DRA) and the photonic crystal (PhC) waveguide. For instance, Withayachumnankul et al. demonstrated a planar dielectric antenna designed using effective medium theory and fabricated on the same PhC slab, achieving a gain of up to 20 dBi³⁰. This approach will enhance the gain of the DRA and hence the maximum detection range. Furthermore, the coupling efficiency between the DRA and the PhC waveguide can be improved through a mode transition, such as a tapered PhC waveguide. Secondly, an algorithm can be developed aimed at improving the detection reliability on the reader side. For example, a calibration-free detection technique based on short-time Fourier transform (STFT) was proposed for chipless RFID tags to improve their detection reliability³¹. These methods will extend the maximum reading range.

References

30. Withayachumnankul, W., Yamada, R., Fujita, M. & Nagatsuma, T. All-dielectric rod antenna array for terahertz communications. *APL Photonics* **3**, (2018).
31. Lin, J. A. et al. Analysis of Calibration-Free Detection Techniques for Frequency-Coded Chipless RFID. *IEEE Trans. Antennas Propag.* **69**, 1681–1691 (2021).

2. Will a higher Q PC cavity improve the sensitivity further of this wireless biosensor? Is the Q factor in this design well-optimized? What are the limitations in getting higher Q in these PCs?

Answer: We thank the reviewer for pointing this out. The sensitivity of this wireless biosensor can be further improved by improving its Q factor. This can be observed in the simulation results in Fig. 3 and the measurement results in Fig. 8 in the manuscript. The total Q factor of the sensor depends on the material Q factor due to material loss, the radiation Q factor due to radiation loss, and the coupling Q factor due to the coupling between the waveguide and the resonator. We have carefully optimized the radiation Q factor by tuning the holes near the resonator. On the other hand, when considering the coupling Q factor, we have decided not to aim for the highest Q factor. The reason is that increasing coupling Q factor leads to a lower coupling efficiency and a lower oscillation signal, which is indicated in the simulation results in Fig. 2c. Consequently, the received signal strength is reduced as shown in the measurement results in Fig. 6d. This leads to a decrease in the reading distance. Therefore, a trade-off between achieving high sensitivity and maintaining a long reading distance has to be made. The limitations in getting higher Q include the compromise between high sensitivity and long reading distance, the structural complexity (the size and the shape of the resonator, time-consuming optimization...), and the loss of the material.

We have updated the manuscript by:

The Q-factor of the sensor depends on material losses, radiation losses, and coupling losses due to the waveguides²⁵. ... The radiation losses can be minimized when a smooth Gaussian distribution of the electric field is achieved along the resonator²⁷. ... The displacement of the holes and the width of the slot are optimized for maximum Q-factor by sweeping the structural dimensions. Further details of design processes can be found in our previous work²³. In addition, the coupling losses depend on the number of hole rows between the waveguide and the resonator, the diameter of the holes, and the number of waveguides used for the coupling. The more rows of holes and the larger the diameter, the better the field confinement within and the lower the

coupling to the resonator. As a result, the Q-factor increases. However, this weakens the transmitted signal. **Since the weakened transmitted signal leads to a reduction of the detected signal at the reader and hence reduces the reading distance, a large number of rows of holes for higher Q factors is not considered in this work.** Here, 3 rows of holes are chosen and different hole radii $r = 0.30p$ and $r = 0.27p$ are compared. The simulated transmission parameter S_{21} is shown in Fig. 2c. The resonant frequencies are 97.440 GHz and 95.144 GHz, with a Q-factor of 2830 and 1740, respectively. As expected, the resonator with larger holes has a higher Q-factor with a lower peak magnitude. **This indicates that the Q-factor should be limited to achieve a reasonable peak magnitude.**

3. The authors have missed some recent developments in THz silicon photonic crystals for sensing/biosensing applications that could be appropriately referenced: APL 121, 011101 (2022); APL 123, 033705 (2023).

Answer: We thank the reviewer for suggesting these recent developments. They have been added to the article.

We have updated the manuscript by:

This has motivated the research of PhC biosensors in the terahertz (THz) frequency range, where PhCs have been studied for wireless communication **and biosensing** applications²⁰⁻²²²⁹.

References

20. Withayachumnankul, W., Fujita, M. & Nagatsuma, T. Integrated Silicon Photonic Crystals Toward Terahertz Communications. *Adv. Opt. Mater.* 6, 1–7 (2018).
21. Kumar, A. et al. Topological sensor on a silicon chip. *Appl. Phys. Lett.* 121, (2022).
22. Navaratna, N., Tan, Y. J., Kumar, A., Gupta, M. & Singh, R. On-chip topological THz biosensors. *Appl. Phys. Lett.* 123, (2023).

Reviewer #2 (Remarks to the Author):

Manuscript: COMMS-23-0344A

Title: 3D Printed Sub-THz Photonic Crystal for Wireless Passive Biosensing

Authors: Y. Zhao et al.

Summary: The authors demonstrate a sub-THz photonic crystal resonator to remotely sense proteins or other bio-molecules. The sensor has a reading range of max 0.9 m and an elevation and azimuth acceptance angle of nearly 90 degrees. It is supposed to be non-electronic, compact, and low-cost.

Comments: The paper reports on an interesting concept for remote sensing of bio-molecules. While the general concept is interesting and original, there are several comments the authors must address prior to publication.

We thank the reviewer for reviewing our manuscript and providing suggestions to improve our work. According to the reviewer's comments, we have responded point-by-point and revised our manuscript as shown in the following. Additions to the text of the manuscript, individual words, and variables that have been replaced or added are highlighted in red.

1. In the abstract the authors claim: "... This will enrich the functionality of the Internet of Things and may help to mitigate pandemics in the future..." I find both claims far-fetched and unsubstantiated. I also don't understand what this has to do with Covid. The only effect the authors have demonstrated is that the resonance peak of a resonator shifts when the index of refraction of the material inside the resonator/slot changes. A well-known effect, nothing new!

Answer: We thank the reviewer for pointing this out. In the first claim we have considered that the proposed remote sensors could be mounted and distributed in the room to monitor its environmental information, such as the concentration of airborne pathogens. This enables a wireless distributed sensor network. It can have various applications in the field of Internet of Things, such as the Internet of Medical Things and environmental monitoring. By using a network of sensors, real-time data on airborne pathogens can be monitored for managing potential health risks.

Regarding the second claim, we have thought that the proposed sensor can be used for pre-infection detection by monitoring the airborne pathogens. In our future work, we plan to functionalize the sensor with specific bio-receptor targeting airborne pathogens, such as corona virus and influenza virus. The distributed wireless sensors in the room can monitor the concentration of the airborne pathogens. If the concentration exceeds a certain threshold, an alarm can be triggered to alert that people may get infected in the monitored room. In this way, the wireless sensors can detect the pathogens in the air at an early state and prevent the spread of the virus efficiently.

We have updated the manuscript by:

The proposed wireless sensor opens the door to a non-electronic, compact, and low-cost solution for distributed remote sensing. ~~This will enrich the functionality of the Internet of Things and may~~

~~help to mitigate pandemics in the future.~~ The remote sensors can be functionalized and extended to a wireless sensor network monitoring airborne pathogen, which may provide a pre-infection detection and help to prevent the spread of viruses efficiently.

2. What is original is that the resonance shift, via a coupler that connects free-space to the resonator, can be detected remotely up to a distance of nearly one meter.

The observed resonance shift, per se, is absolutely unspecific and reveals no information whatsoever on the sample other than the magnitude of the change in index of refraction. Whether this is caused by water, salt, sugar or some protein can not be inferred from this measurement. At some point the authors claim that specificity comes into the game when the surface is functionalized so that only very specific bio-molecules can bind to it. This would result in a more or less mono-molecular layer of bio-molecules or proteins etc. and the setup - as presented - is not sensitive enough to measure such small modifications to the index of refraction. Recall the minimum layer thickness required to observe a detectable resonance shift is 250 nm. Hence, claim and reality do not match and require further clarification. I would find it mandatory to repeat the measurement with a functionalized surface/slot and to substantiate the claims with a realistic measurement. Otherwise the claims have to be reformulated reflecting the realities of the presented results.

Answer: We thank the reviewer for raising this question. We agree with the reviewer that the current design may not have the required sensitivity to detect monolayer biomolecules or proteins. The specificity of the resonant shift would come from the functionalized specific bio-receptor. The proposed concept needs to be further improved. In the current work, it is stated that the resulting acceptable confidence is 250 nm considering 3 times the standard deviation. Currently, we are working on improving the sensitivity by increasing resonant frequency, increasing the covered area of the analyte on the sensor, and lowering measurement tolerances.

(a) The first approach is increasing the resonant frequency to improve the sensitivity. To demonstrate its potential benefits, we have conducted simulations where the resonator is scaled to different resonant frequencies ranging from 100 GHz to 1000 GHz. A thin film of analyte with a constant refractive index (1.8) and a constant thickness (0.1 μm) is applied to the wall of the slot in the resonator. Their resonance shifts and FOMs are simulated and plotted in the following figure. It shows that as the resonant frequency increases, the resonance shift increases significantly and the sensitivity is improved.

(b) In the current work, the analyte is deposited on the wall of the slot to demonstrate the principle. In the future work, it is planned that the analyte covers the complete surface of the sensor, including the walls of the slot, the walls of all the holes, and the top and bottom surface of the sensor. More coverage area leads to higher resonant shift and higher sensitivity. Simulations of the resonator with the analyte covering different positions and areas have been conducted and added to the Supplementary Information.

(c) In our measurement, the detectable shift is defined by considering 3 times the standard deviation. The variations in temperature in the room and device noise contribute to measurement tolerances. In our previous work (doi:10.1109/ACCESS.2022.3202537), we have introduced a reference resonator to minimize these tolerances. The difference between the resonant frequencies of the reference resonator and the resonator for sensing remains almost constant with varying temperature. This approach can help minimize the standard deviation and decrease the minimum detectable layer thickness.

Based on these considerations, the sensor can be improved for monolayer detection in the future work. Based on the reviewer's comment, we have reformulated the claims.

We have updated the manuscript by:

If the resonance shift must be higher than 3 times the standard deviation to achieve a 99.73% possibility of correlation, the resulting acceptable confidence is 1.8 μg or 250 nm thick biomolecule film. An essential part of the measurement tolerance relates to variations in ambient temperature and device noise. Temperature fluctuations can lead to resonant shifts. As presented in our previous work²³, the influence of the environment can be minimized by introducing a reference resonator. The difference between the resonant frequencies of the reference resonator and the resonator for sensing remains almost constant with varying temperature. This approach can effectively minimize the standard deviation enabling a decrease in the minimum detectable layer thickness. Furthermore, the sensitivity can be further improved by increasing resonant frequency²⁸.

...

Furthermore, in future work, the sensitivity of the current design should be improved to achieve a reliable detection of a monolayer of biomolecules and proteins. To achieve this, its resonant frequency can be increased and the analyte can be deposited on a larger area of the sensor, including the walls of all the holes and the top/bottom surface of the resonator. This will also

facilitate the immobilization process in the next steps. Detailed studies on the effects of the resonant frequency and the covered area can be found in Supplementary Material Note 5 and 6.

We have updated the Supplementary information by:

5. Effect of the resonant frequency on the resonant shift and FOM

The resonance shift due to the analyte depends on the resonant frequency. To study this, the resonator with two ports is scaled to different frequencies ranging from 100 GHz to 1000 GHz and simulated in CST. A thin film of analyte with a constant refractive index (1.8) and a constant thickness ($0.1 \mu\text{m}$) is deposited on the walls of the slot in the resonator. Their resonance shifts and FOMs are simulated and plotted in the following figure. The results indicate that as the resonant frequency increase, the resonance shift increases greatly and the sensitivity is enhanced.

Figure 3. Resonant shift of the resonators with different resonant frequencies with a $0.1 \mu\text{m}$ thin film as the analyte.

6. Effect of the analyte covering different areas on the sensor

In this manuscript, the analyte is deposited on the walls of the slot as a proof of principle. However, it is important to note that a larger covered area by the analyte leads to stronger interaction between electromagnetic wave and the analyte. As a result, the resonant shift due to the analyte is increased and the sensitivity is improved. To investigate the effect of different positions, the resonator is simulated with the analyte positioned on different areas in CST. The different positions of the analyte and the simulated resonant shift due to it are presented in Fig. 4-9.

Figure 4. The analyte (in red) covers the wall of the slot (area 1), which is the same as in the manuscript. The simulated resonant shift is 24 MHz.

Figure 5. The analyte (in red) covers the wall of the holes around the slot (area 2). The simulated resonant shift is 14 MHz.

Figure 6. The analyte (in red) covers the wall of the holes of the whole sensor (area 3). The simulated resonant shift is 18 MHz.

Figure 7. The analyte (in red) covers the top and bottom surface around the slot (area 4). The simulated resonant shift is 10 MHz.

Figure 8. The analyte (in red) covers the top and bottom surface in a large area of the resonator (area 5). The simulated resonant shift is 14 MHz.

Figure 9. The analyte (in red) covers the top and bottom surface of the whole resonator (area 6). The simulated resonant shift is 14 MHz.

3. What would happen if the voids of the photonic crystal also fill with the same pathogen? A shift in band gap? What would be the consequence? Have the authors tested - via simulation - what happens if the bio-molecules not only fill the slot but also the voids between the pillars of the photonic crystal structure?

Answer: We thank the reviewer pointing this out. If the thin-film analyte is deposited on the wall of the holes, similar as on the wall of the slot, the resonant frequency shifts towards the lower frequency. The value of the resonance shift depends on the interaction between the analyte and the electric field. Based on the electric field distribution, the resonator has the maximum field concentration in the slot. Consequently, the analyte positioned in the slot results in the maximum shift. On the other hand, the top/bottom surface of the slab or the wall of the holes far away from the slot experience lower electric field strength. As a result, the positioned analyte deposited on those surfaces has a smaller effect on the resonance shift. However, covering all the surface of the sensor with the analyte can increase the overall resonance shift and the sensitivity. In our current measurement setup, the analyte is deposited only on the walls of the slot as a straightforward proof of principle. Now, we have simulated the resonator with the analyte on different sites and summarized the results in the Supplementary information. The updated part in Supplementary information can be found in the response to the second comment of the reviewer.

4. Moreover, the remote sensing aspect requires clarification. Suppose the device is functionalized for detection of some specific pathogen and at some point in time will be fully covered, i.e. saturated. Then the device becomes useless unless cleaned. This clearly limits the use of the detector to some very specific applications, which the authors should elaborate on. Is the remote sensing aspect really that important then? Time stamp information is directly linked to the read-out frequency.

Answer: We thank the reviewer for raising this out. We agree with the reviewer that the biosensors will face the issue of fully covered surfaces, making them unusable for further detection. This is a general issue for various biosensors. In such cases, there are typically two solutions: (a) Clean the sensor at the end of a sequence by rinsing it with a specific solution or (b) the sensors are designed for single use and are disposed after reaching saturation. However, solution (a) introduces additional cost and complex procedures. Our sensor is designed to be non-electronic, passive, and low-cost. Hence, it can be easily disposed after a single use.

To effectively monitor airborne pathogens, multiple biosensors must be distributed across an area. If the sensors cannot be read remotely, two potential methods can be used: (a) using numerous

cables to connect each sensor to a central reader and (b) physically moving the reader to contact each sensor for reading. Both methods are complex, time-consuming, and impractical for large-scale deployments. In contrast, the remote sensors can be wirelessly read in a large distance and with a large acceptance angle for an easy use.

We have updated the manuscript by:

The measurement results show a long reading range, a large reading angle, and good wireless sensitivity for a model BSA protein. **This enables the distribution of remote sensors in a large area, i.e. a room or hallway, for monitoring airborne pathogens. In comparison to sensors connected to a central reader with cables or those requiring reading in proximity, the proposed remote sensor offers easy and efficient reading capabilities, making it highly suitable for large-scale deployments.** In addition, the proposed sensor can operate in critical environments without an active chip and battery and has a long lifetime.

...

With the advantage of low cost fabrication, the wireless sensor can be disposed once the pathogen is captured and detected. This eliminates the need for complex cleaning processes for reuse.

5. Further, I would expect some more details on the simulations and the material parameters used.

Answer: We thank the reviewer for raising this out. Now more details on the simulation and the material parameter are added in the Supplementary Information.

We have updated the Supplementary information by:

7. More details of simulation setup

To optimize the sensor and analyze its sensitivity, the transmission parameter of the resonator with two tapers is simulated with CST as shown in Fig. 10. Both tapers are inserted into a WR10 rectangular waveguide (shown in yellow transparent) for excitation and receiving. Waveguide ports are used for the WR10 waveguide. Because the expected frequency spectrum has a resonance peak with high Q factor, a frequency domain solver is chosen to reduce simulation time. When using a time domain solver, the simulating time of a high-Q resonator is very long and the simulated Q factor depends on the simulation time. Since the radiation pattern of the resonator is not needed, an open background is used. In the solver setup, a single frequency sample is set near the resonant frequency and automatic frequency samples are chosen. Furthermore, adaptive mesh refinement is activated to improve accuracy. The material of the resonator is defined as: Epsilon=9 and Tangent delta=0.00022. The dimensional parameters such as radius of holes, slot width, and shift of the holes are studied by a parameter sweep.

Figure 10. The model of the resonator and the simulation setup in CST.

Reviewer #3 (Remarks to the Author):

The authors propose and demonstrate an interesting RI biosensing concept with a long stand-off distance.

The device uses a PhC designed to operate in the W band (75-110GHz) regime. The authors also 3D fabricate a structure and demonstrate a proof-of-concept device.

We thank the reviewer for reviewing our manuscript and providing suggestions to improve our work. According to the reviewer's comments, we have responded point-by-point and revised our manuscript as shown in the following. Additions to the text of the manuscript, individual words, and variables that have been replaced or added are highlighted in red.

While interesting, this work seems better suited to a biosensors or a devices journal as the novelty in the communication strategy and its fit for this journal is not clear.

Answer: We thank the reviewer for raising this point. At the beginning, this manuscript has been submitted to the Journal Nature Communications. The editor of Nature Communications has suggested to us to transfer/resubmit this manuscript to Communications Engineering. We think the reason why the editor has suggested Communications Engineering is that the concept and the sensor are in the focus and designed from the view of engineers and that the work can be considered as a proof-of-concept.

If the authors choose to modify and resubmit the manuscript either here or to another journal, it would benefit the authors to clarify the following observations to further strengthen their manuscript:

(a) the sensor assumes a clear line-of-sight between the sensor and the detector. However, in real-world applications, there are obstructions (including human subjects and walls) between the sensor and the detector, specifically for the proposed application. The presence of such obstacles has shown to severely attenuate W-band signals. How can such a device be implemented, in say, crowded areas? Why would it be disadvantageous to detect a signal in close proximity and wirelessly transmit the electrical signal given IoT promises to make devices cheap and connected?

Answer: We thank the reviewer for raising this point. In our concept, the oscillating signal of the resonator is extracted using a time gate to remove reflections from obstacles near the sensor, such as the table on which the setup is located. As the reviewer said, the obstacles within the line-of-sight between the sensor and the reader severely attenuate W-band signals, which is a general issue for wireless communication at these frequencies. A straightforward solution for the proposed concept is to move the reader (e.g. using a robot) around the monitored room. Since the remote sensors have a long reading distance and a wide reading acceptance angle, the moving reader can easily detect the remote sensors. Furthermore, the distribution of many remote sensors in a large area must be organized in a way that they can be read easily. It is not mandatory to read all the sensors or get the signal all the time to monitor the room.

If we understand the comment correctly, the reviewer considers a solution, which is integrated with a module for close and wireless communication, such as an NFC module or IoT module. However, they face some practical challenges. Firstly, to detect these sensors in close proximity,

people must bring the reader device close to all distributed sensors within a given area. When dealing with a large number of sensors, this procedure leads to a time-consuming work. It is even more difficult in a crowded room. Secondly, NFC requires the use of complex circuits, including energy-harvesting, frequency convertor, sensor itself and more. This makes the sensor bulky, complex, and expensive. Proper disposal of such circuits, especially those containing batteries, must be carefully considered. We think using electromagnetic waves for both communication and sensing makes the sensor chip compact, low cost, efficient, and easily disposable.

We have updated the manuscript by:

However, the measurement results with varying ϕ and θ show that the wireless sensor can be read over a wide angle of incidence. **In the measurement setup, a line-of-sight between the sensor and the antenna is used. Obstacles within the line-of-sight may attenuate the signal severely and affect the detection. However, thanks to the large reading range and the large acceptance angle of the sensor, this problem can be solved by simply moving the reader.**

...

The measurement results show a long reading range, a large reading angle, and good wireless sensitivity for a model BSA protein. **This enables the distribution of remote sensors in a large area, i.e. a room or hallway, for monitoring airborne pathogens. In comparison to sensors connected to a central reader with cables or those requiring reading in proximity, the proposed remote sensor offers easy and efficient reading capabilities, making it highly suitable for large-scale deployments.** In addition, the proposed sensor can operate in critical environments without an active chip and battery and has a long lifetime. **The non-electronic structure is compact, low-cost, and hence easily disposable.**

(b) as opposed to structures such as RFIDs that can be produced in the millions in a short amount of time at a very-low cost, the proposed 3D manufacturing approach will only be able to fabricate a low volume of devices irrespective of the claim made by the authors in terms of production speed (taking several hours for curing itself). The 3D manufacturing approach just cannot match screen-printing, for example in terms of throughput.

Answer: We thank the reviewer for raising this point. We agree with the reviewer that the 3D printing technique is typically suitable for low-volume production of devices. In our work, we have chosen 3D manufacturing for rapid prototyping purposes, aiming to quickly fabricate several samples to prove the principle. However, it is important to note that the proposed photonic crystal structure can be fabricated using micro- and nanofabrication techniques. In our previous work, we successfully utilized a standard process called deep reactive ion etching (DRIE) to etch high resistance silicon wafers and realize the desired photonic crystal. However, from layouting, ordering photo masks, waiting for fabrication, to trying different designs, it has taken several months to complete. In contrast, the 3D printing fabrication enabled to produce the prototypes within a few weeks, allowing for faster iteration and evaluation of multiple samples. Nevertheless, it has been demonstrated that the proposed structure can be massively produced using DRIE for a high throughput.

We have updated the manuscript by:

To enable rapid prototyping and have maximum freedom in the design, a 3D printing process is used to fabricate the samples. **Compared with micro- and nanofabrication, the 3D printing process has lower cost and a shorter fabrication time for a small volume of samples. However, large volumes can be fabricated very cost-efficiently with standard silicon processes as demonstrated in our previous work²⁸.**

References

28. Zhao, Y. et al. Photonic Crystal Resonator in the Millimeter / Terahertz Range as a Thin Film Sensor for Future Biosensor Applications. *J. Infrared, Millimeter, Terahertz Waves* (2022) doi:10.1007/s10762-022-00859-1.

(c) based on the above facts, the authors should comment on the potential deployability of millions of devices and cost of manufacturing.

Answer: We thank the reviewer for raising this point. As mentioned above, the proposed sensor can be fabricated on a silicon wafer using standard DRIE process. The proposed photonic crystal structure has periodic holes and a slot with the same depth. Using a patterned mask, the holes and the slot can be etched simultaneously and no additional complex steps are required. In addition, as the resonant frequency increases, the sensor can be scaled down in size. This means that more sensors can be produced from the same wafer. This increases the throughput for each wafer and hence of the fabrication process. Furthermore, the smaller size of the sensors corresponds to a reduced depth of the holes, resulting in shorter etching time. By using micro- and nanofabrication techniques, the proposed sensor has the potential for cost-effective and massive production. However, a higher frequency requires a more efficient design as the FSPL etc. increases. This must be addressed in future work

We have updated the manuscript by:

Considering the future potential, the sensor can be manufactured in large quantities on high resistance silicon wafers (HRSi) using a standard process such as deep reactive ion etching (DRIE) to reduce costs and increase throughput. The throughput per wafer can be further increased by increasing the resonant frequency as this reduces the size of the sensor.

(d) the authors point out energy harvesting and signal processing as some of the disadvantages for the current state-of-the-art electronic sensors. One can also argue that deploying a network analyzer or a detector in the W-band comes with significant investments and operation know-how, plus the added difficulty of aligning the detector and the sensor structure with high precision (that is one detector per a few sensors, which is not practical). Also, how would this system multiplex signals from multiple sensors without deploying complex electronic processing equipment at the detector or the sensor head?

Answer: We thank the reviewer for raising this out. At this stage, we are primarily focused on the proof of the principle of our concept. Realizing a low-cost device will be the next step of our work. To achieve this, we will adopt a radar system, considering existing commercial radar products in the W-band and in the WR3 (220 – 330 GHz) band. By using existing technology, we can reduce cost of the reader. Regarding the user interaction, a graphical user interface can be developed if

we have the chance to transfer this concept into a product in the future. Furthermore, because the sensor works in far field over a relatively long distance and a wide acceptance angle, the sensor can be easily read without precise alignment.

We agree with the reviewer that multiple sensors monitoring environment need to be distributed in different positions in the room and multiplexing signal is required. To address this, we have considered multiplexing the signals in different frequencies. By tuning the resonator to different frequencies, we can differentiate the sensors on different positions. On the reader side, the frequency spectrum is analyzed and the positions of different resonant frequencies can be related to different sensor positions. Searching for the resonant frequencies can be performed through simple data processing. Extra electronic equipment is not needed. This simplifies the reader design, reducing its cost, and do not need encoder and memory, which are typically required for identification in electronic sensors.

We have updated the manuscript by:

Furthermore, the proposed concept can be extended with a simplified radar system serving as the reader and multiple sensors tuned to different resonant frequencies for multiplexing their signals and identify their locations³²⁻³⁴.

References

32. Sene, B., Reiter, D., Knapp, H. & Pohl, N. Design of a Cost-Efficient Monostatic Radar Sensor with Antenna on Chip and Lens in Package. *IEEE Trans. Microw. Theory Tech.* 70, 502–512 (2022).
33. Sanchez-Pastor, J. et al. Evaluation of Chipless RFID Indoor Landmarks at 80 GHz and 240 GHz Using FMCW Radars. 2022 16th Eur. Conf. Antennas Propagation, EuCAP 2022 (2022) doi:10.23919/eucap53622.2022.9768965.
34. Burmeister, T. et al. Chipless frequency-coded RFID tags integrating high-Q resonators and dielectric rod antennas. 15th Eur. Conf. Antennas Propagation, EuCAP 2021 (2021) doi:10.23919/EuCAP51087.2021.9410915.

(e) PhC structures are very sensitive to any small changes in structural parameters or any changes in the environment. How does change in ambient temperature/humidity affect the performance of the devices?

Answer: We thank the reviewer for raising this question. We agree with the reviewer that PhC resonators are sensitive to changes in structural parameters or changes in the environment. A structural change in the area, where the analyte is deposited, affects its resonant frequency. The change in environment can affect the position and the magnitude of the resonant peak. In particular, the change in temperature can affect the permittivity and loss tangent of the material used. According to measurement results in our previous work (doi: 10.23919/EuCAP48036.2020.9135861), the resonant frequency and Q factor change with varying ambient temperature. To address this challenge, we have developed a sensor consisting two channels in our previous work (doi: 10.1109/ACCESS.2022.3202537). One channel works as the sensing channel for sensing analyte, while the other channel works as a reference channel without the presence of the analyte. Their resonant frequencies react with the environment in the same manner. As a result, the difference of their resonant frequencies remains almost constant with

varying temperature. This approach help mitigate the impact of the temperature. Regarding ambient humidity, we have not observed any significant effect of the sensor. The change in ambient humidity may result in a change of the permittivity and loss of the air within the holes and the slot. However, the sensor consisting of a reference channel and a sensing channel is able to effectively eliminate the influence of the environment.

We have updated the manuscript by:

If the resonance shift must be higher than 3 times the standard deviation to achieve a 99.73% possibility of correlation, the resulting acceptable confidence is 1.8 μg or 250 nm thick biomolecule film. **An essential part of the measurement tolerance relates to variations in ambient temperature and device noise. Temperature fluctuations can lead to resonant shifts. As presented in our previous work²³, the influence of the environment can be minimized by introducing a reference resonator. The difference between the resonant frequencies of the reference resonator and the resonator for sensing remains almost constant with varying temperature. This approach can effectively minimize the standard deviation enabling a decrease in the minimum detectable layer thickness.**

References

23. Zhao, Y. et al. Sensitive and Robust Millimeter-Wave/Terahertz Photonic Crystal Chip for Biosensing Applications. *IEEE Access* 10, 92237–92248 (2022).

(f) a stand-off distance of 0.5m is still not sufficient, especially given the transmissivity and diffusivity of viruses in aerosol form.

Answer: We thank the reviewer for pointing this out. In our manuscript, we have developed a proof-of-principle remote sensor prototype. Our current focus is on enhancing the detected signal to increase the reading distance. We would like to address this concern by discussing two potential approaches:

(a) The first method is that the received signal can be enhanced by increasing the gain of the dielectric rod antenna (DRA) and optimizing the coupling between the photonic crystal (PhC) waveguide and the DRA. In Fig. 6d, we observe that the sensor with a higher gain (about 5 dB) shows a higher magnitude (about -70 dB at 0.5 m), indicating that increasing antenna gain enhances the detected signal. According to the literature (doi: 10.1063/1.5060631), a planar dielectric antenna fabricated on the same photonic crystal slab can be designed using effective medium theory to reach a gain of 20 dBi. Hence, our current DRA has the potential for a 15 dB improvement. Additionally, the coupling between the PhC waveguide and the DRA can be improved through a mode transition, such as a tapered PhC waveguide.

(b) Secondly, we are studying a data-processing algorithm to enhance detection reliability. In Fig. 6c, the resonant peak for a 0.5 m distance is about 15 dB higher than the noise level. The resonant frequency can be identified by searching the maximum peak. For a long distance with a low signal-noise-ratio, the resonant signal can be enhanced and the noise can be reduced. In the literature (doi: 10.1109/TAP.2020.3016160), Lin *et al.* developed a method based on short-time Fourier transform (STFT), averaging, and filtering to detect chipless RFID tags. This approach can potentially be applied to improve the detection reliability in our proposed sensor.

In summary, the current reading distance achieved by our proposed sensor is longer than the state-of-the-art and can be further improved by optimizing the sensor's structure and developing suitable algorithms.

We have updated the Discussion of the manuscript by:

Our future effort will be dedicated to improving the performance of the proposed sensor and considering its practical applicability. Firstly, the detected signal can be enhanced by employing a high-gain antenna and optimizing the coupling efficiency between the dielectric rod antenna (DRA) and the photonic crystal (PhC) waveguide. For instance, Withayachumnankul et al. demonstrated a planar dielectric antenna designed using effective medium theory and fabricated on the same PhC slab, achieving a gain of up to 20 dBi³⁰. This approach will enhance the gain of the DRA and hence the maximum detection range. Furthermore, the coupling efficiency between the DRA and the PhC waveguide can be improved through a mode transition, such as a tapered PhC waveguide. Secondly, an algorithm can be developed aimed at improving the detection reliability on the reader side. For example, a calibration-free detection technique based on short-time Fourier transform (STFT) was proposed for chipless RFID tags to improve their detection reliability³¹. These methods will extend the maximum reading range.

References

30. Withayachumnankul, W., Yamada, R., Fujita, M. & Nagatsuma, T. All-dielectric rod antenna array for terahertz communications. *APL Photonics* **3**, (2018).
31. Lin, J. A. *et al.* Analysis of Calibration-Free Detection Techniques for Frequency-Coded Chipless RFID. *IEEE Trans. Antennas Propag.* **69**, 1681–1691 (2021).

Summary: while the work tries to showcase a biosensing concept using a high-freq PhC structure and claims to achieve a large sensing distance, the complex fabrication process (slow speed, low repeatability, high cost) and practical constraints in deploying hundreds and thousands of such sensors make them less effective or attractive compared to the state-of-the-art devices.

Answer: We thank the reviewer to raise this practical question. As explained for the reviewer's comment (b) and (c), the proposed sensor based on PhC can be fabricated on a silicon wafer using standard DRIE process. It enables massive production with low cost and high volume. The proposed remote sensor without electronics and battery has a long reading distance, which will be a powerful competitor for the state-of-the-art devices.

Reviewers' comments:

Reviewer #1 (Remarks to the Author):

The authors have responded well to all the three reviewer comments and I think the manuscript could now be accepted for publication.

Reviewer #2 (Remarks to the Author):

The authors have answered all my comments and have changed the text accordingly. I am fully satisfied with the modifications.

Reviewer #3 (Remarks to the Author):

I think the authors for taking the time to address my concerns. However, the suggested answers have still not completely or satisfactorily answered the questions.

1. The large crowd areas is still a concern and moving the sensor around with a robot does not seem to be a practical or a well conceived solution.
2. The authors suggest that NFC and others would be complex due to complex read out, but given that time gate is proposed as a solution, that is a fairly complex solution as well and no easier than other methods. In fact, it could be much more expensive and more complex at those frequencies.
3. The fabrication methodology for DRIE is super complicated (deep trenches not being perfectly vertical and being jagged. It is time consuming since dry etching is slow as you go deeper. The authors say it has been demonstrated that DRIE is a rapid solution, but compared to other fabrication procedures, it is ver complex and expensive.
4. The biosensing aspect still doesn't fit the journal scope. The authors need to consider another suitable journal. The proof-of-concept explanation does not satisf the condition to be considered in this journal.

Reviewer #3 (Remarks to the Author):

I think the authors for taking the time to address my concerns. However, the suggested answers have still not completely or satisfactorily answered the questions.

We thank the reviewer for reviewing our manuscript again and providing suggestions to improve our work. According to the reviewer's comments, we have responded point-by-point and revised our manuscript as shown in the following. Additions to the text of the manuscript, individual words, and variables that have been replaced or added are highlighted in red.

(1) The large crowd areas is still a concern and moving the sensor around with a robot does not seem to be a practical or a well conceived solution.

Answer: We thank the reviewer for raising this point. The reviewer has the concern that “moving the **sensor** around with a robot does not seem to be a practical or a well conceived solution”. We agree with the reviewer that moving the sensor around a crowded area is not practical. In fact, in our previous response to the reviewer, we have proposed two possible strategies: (1) moving the **reader** around and (2) carefully planning the placement of the sensors to ensure an effective operation. It is more feasible to move only the reader to read each sensor distributed in the area. The movement of the reader can be accomplished by either a robot or a person, depending on the specific requirements and constraints of the monitored environment. On the other hand, we would like to emphasize that it is not mandatory to read all the sensors or get the signal all the time to monitor the room. By carefully organizing the distribution of the sensors, most areas can be covered to effectively monitor the spread of airborne pathogens.

We have updated the manuscript by:

Obstacles within the line-of-sight may attenuate the signal severely and affect the detection. **However, the large reading range and acceptance angle of the sensor allow for a solution to this problem by moving the reader to read every sensor distributed in the monitoring area. Additionally, not every sensor needs to be read at all times. However, thanks to the large reading range and the large acceptance angle of the sensor, this problem can be solved by simply moving the reader.**

(2) The authors suggest that NFC and others would be complex due to complex read out, but given that time gate is proposed as a solution, that is a fairly complex solution as well and no easier than other methods. In fact, it could be much more expensive and more complex at those frequencies.

Answer: We appreciate the reviewer for bringing up this point. As previously discussed, a wireless sensor integrated with an NFC module, an IoT module, or other solutions has complex circuits, including energy harvesting, frequency conversion, and the sensor itself. It is important to note that this complexity exists at the remote sensor, which can be considered a disadvantage in this regard. In our concept, the remote sensor consists solely of the passive dielectric sensor and antenna, without any additional circuit or battery. This leads to significant simplification and cost reduction. The circuits for signal reading and processing will be integrated into the reader, including the implementation of time gating. The signal processing, including time gating, can be easily achieved using a portable and low-cost Raspberry Pi, a small single-board computer.

We have updated the manuscript by:

In addition, the proposed sensor can operate in critical environments without an active chip and battery and has a long lifetime. ~~The passive remote sensor is non-electronic, compact, and inexpensive, making it easy to dispose of. This gives it an advantage over solutions that use active circuits. The non-electronic structure is compact, low-cost, and hence easily disposable.~~

...

Secondly, an algorithm can be developed aimed at improving the detection reliability ~~on a low-cost, small single-board computer~~ on the reader side.

(3) The fabrication methodology for DRIE is super complicated (deep trenches not being perfectly vertical and being jagged). It is time consuming since dry etching is slow as you go deeper. The authors say it has been demonstrated that DRIE is a rapid solution, but compared to other fabrication procedures, it is ver complex and expensive.

Answer: We appreciate the reviewer for bringing up this question. In our previous response, we explained that the suggested structure can be produced on a large scale using deep reactive-ion etching (DRIE) for high throughput. However, DRIE fabrication is time-consuming. On the other hand, 3D printing allows for rapid prototyping and low-volume production of the sensors.

Deep Reactive Ion Etching (DRIE) is a widely used process for creating high aspect ratio holes and trenches in wafers and substrates. The process involves alternating etching and passivation steps using reactive ions to remove material from the substrate. This process is commonly referred to as the Bosch process. During the passivation phase, a polymer protective layer is deposited on the sidewalls of the etched holes to prevent lateral etching and achieve high aspect ratios. The fabrication process of this material presents certain difficulties. Firstly, etch rates may decrease as aspect ratios increase due to the challenge of getting etchant into narrow and deep holes. Secondly, the cyclic process can result in multiple, rough, and semicircular steps on the sidewalls of the etched holes. Finally, potential non-uniformities can cause imperfectly vertical sidewalls.

To address these challenges, optimizing process parameters such as gas flow rates, plasma power, etch and passivation cycle times, and chemistry is often necessary. Additionally, designing features to be etched can help mitigate these effects. For instance, using high-density plasma can improve etch rate, and fine-tuning the duration of etching and passivation steps can minimize step size. Additionally, etch masks and patterns can be designed carefully to distribute etching forces evenly. The optimization of multiple process parameters can address the challenges.

The proposed PhC structure may have non-perfect sidewalls of the holes due to the mentioned fabrication difficulties. The sidewall may be slightly tilted or have multiple rough and semicircular steps, both of which can cause a deviation in the resonant frequency of the PhC resonator. In a separate project, we characterized nine DRIE-etched 250 GHz PhC resonators randomly selected from the same wafer. The standard deviation of their resonant frequencies was only 0.1% of the average resonant frequency. This result was obtained even before optimizing the DRIE process parameters, indicating that the variation is minor compared to the resonant frequency and can be further reduced. Furthermore, the required aspect ratio in our case is relatively low. The resonator has a thickness of 550 μm and the slot width (which is the finest feature) is 244.5 μm , resulting in

an aspect ratio of 2.2. This aspect ratio is not particularly challenging for the DRIE process, which can achieve an aspect ratio of 30 or higher. Scaling the operation frequency does not significantly affect the aspect ratio, as higher frequencies require smaller features as well as thinner substrates.

It is important to note that absolute uniformity in resonant frequency is not critical for our application. As previously mentioned, the resonator can be extended to two channels to compensate for environmental effects. The difference in their resonant frequencies can be recorded before deployment, and any subsequent changes indicate the captured pathogen. This method is not affected by initial frequency deviations.

We have updated the manuscript by:

Considering the future potential, the sensor can be manufactured in large quantities on high resistance silicon wafers (HRSi) using a standard process such as deep reactive ion etching (DRIE) to reduce costs and increase throughput. **Using DRIE, it is possible to achieve an aspect ratio of 30 or more, while the proposed sensor only has an aspect ratio of 2.2. This method can be easily transferred to PhC resonators. In preliminary investigations, we characterized 9 DRIE-etched 250-GHz PhC resonators randomly selected from the same wafer. The standard deviation of their resonant frequencies was only 0.1% of the average resonant frequency. This result was obtained without optimizing the DRIE process parameters.**

(4) The biosensing aspect still doesn't fit the journal scope. The authors need to consider another suitable journal. The proof-of-concept explanation does not satisfy the condition to be considered in this journal.

Answer: We thank the reviewer for this concern. As we mentioned in the previous response, the editor of Journal Nature Communications has suggested us to transfer/resubmit this manuscript to Communications Engineering. And the editor of Communications Engineering has confirmed that our paper is suitable for Communications Engineering.